# Association of Sarcopenia with Osteopenia and Osteoporosis in Community-Dwelling Older Korean Adults: A Cross-Sectional Study

**DOI:** 10.3390/jcm11010129

**Published:** 2021-12-27

**Authors:** Do-Youn Lee, Sunghoon Shin

**Affiliations:** 1Research Institute of Human Ecology, Yeungnam University, Gyeongsan-si 38541, Gyungbuk, Korea; triptoyoun@yu.ac.kr; 2Neuromuscular Control Laboratory, Yeungnam University, Gyeongsan-si 38541, Gyeongbuk, Korea

**Keywords:** sarcopenia, osteopenia, osteoporosis, elderly

## Abstract

Sarcopenia and bone disorders, such as osteopenia and osteoporosis, are common musculoskeletal disorders in older adults. Therefore, this study aimed to establish the association between sarcopenia and bone disorders such as osteoporosis and osteopenia according to sex. We analyzed 3077 participants from the 2008–2011 Korean National Health and Nutrition Examination Survey aged 65 years or older. After adjusting for all covariates, such as physical examinations, exercise, and nutrient intake (model 4), the odds ratios for the association between sarcopenia and bone disorders were 2.051 (95% confidence interval [CI]: 1.498–2.808) in osteopenia and 2.258 (95% CI: 1.584–3.218) in osteoporosis. However, when sex was analyzed separately, the odds ratio was significantly different in men (osteopenia—2.068, 95% CI: 1.462–2.924; osteoporosis—3.247, 95% CI: 1.953–5.399), but not in women. Therefore, the results of this study show an association between sarcopenia and bone disorders in older Korean adults. Sarcopenia is significantly related to osteopenia and osteoporosis, especially in men, when stratified by sex.

## 1. Introduction

Age-related sarcopenia and osteoporosis are the most common musculoskeletal disorders [1]. Decreases in skeletal muscle and bone mass associated with aging have consequences such as falls, poor balance, fractures, and frailty [2,3]. Sarcopenia is a term that was initially used to describe skeletal muscle wasting caused by aging [4], which is accompanied by metabolic and endocrine dysfunction and is especially related to an increase in tumor necrosis factors (TNF) [5]. Osteoporosis is a metabolic bone disease [6], and these two diseases are associated with an increase in TNF in the context of chronic inflammatory diseases [7].

Although its mechanisms have not yet been identified, sarcopenia is defined as a decrease in muscle mass and muscle strength due to changes in numerous body components triggered by aging, lack of exercise, and hormonal imbalance [8,9,10,11]. More than 50 million people worldwide developed sarcopenia in 2000; this number is expected to increase to more than 200 million by 2040 [12]. As of 2014, the prevalence of sarcopenia in Asian countries was 5.1–21.0% in men and 4.1–16.3% in women [10]. In particular, the prevalence of sarcopenia in Korea was 18.4% in people aged 70 years or older as of 2017 [3].

Osteopenia and osteoporosis are diseases in which the bone strength decreases due to reduced bone density, which increases the risk of fracture [13]. These two diseases caused by bone structure deterioration and bone degeneration are referred to as bone disorders [14]. Bone strength reflects the following two main characteristics: bone density is determined by bone mass and loss, while bone quality is determined by bone turnover and remodeling [15]. In addition, the prevalence of bone disorder disease increases with age [16].

Both sarcopenia and osteoporosis are caused by several factors, and some of the factors that cause osteoporosis also trigger sarcopenia [17]. In particular, bone and muscle tissues are closely linked and affected by endocrine and paracrine factors [18]. In addition, aging, inactivity, and environmental factors are known to simultaneously cause negative changes in muscle and bone mass [19,20].

Sarcopenia and osteoporosis can lead to secondary diseases such as falls, fractures, and functional damage [21]. Recent meta-analysis studies have reported that patients with sarcopenia have a higher risk of falls than the elderly with non-sarcopenia [22]. The high risk of falls leads to fractures in the elderly [23], which causes high disease burden costs at the national level [24]. In addition, the two diseases are also deeply related to the nutritional aspects of the elderly [25]. Although aging is the most common risk factor for both diseases, the incidence of sarcopenia has been found to be associated with malnutrition [26], and the prevalence of malnutrition is also high in hospitalized fracture patients caused by osteoporosis [27,28]. These results imply that the prevention and therapeutic management of sarcopenia and osteoporosis is important and necessary [29].

As such, sarcopenia and bone disorders are common diseases caused by aging, but it remains unclear whether they should be considered as one disease or two different diseases [30]. Bone disorder is often thought of as a female disease due to the hormonal changes occurring during the menopausal period as age increases [31]. Also, the prevalence of osteoporosis in men is lower than in women, which is largely due to greater bone size, bone mass, and shorter lifespan [32]. However, osteoporosis is becoming an increasingly important problem in men because the mortality rate after fracture from osteoporosis is also high in men [33,34].

Preliminary studies on the association between sarcopenia and osteoporosis were either only conducted in women [35,36], or the variables that could affect the two diseases, such as lifestyle, exercise status, and nutrition factors, were not controlled [37,38]. Moreover, most studies have not analyzed the association by classifying bone disorders such as osteoporosis and osteopenia based on the degree of bone mineral density (BMD) and did not identify the association between the two diseases according to sex [36,37,38]. To this end, this study aimed to establish the association between sarcopenia and bone disorders such as osteoporosis and osteopenia according to sex using the data from the Korean National Health and Nutrition Examination Survey (KNHANES).

## 2. Materials and Methods

### 2.1. Data Source and Sampling

This study used the data from KNHANES (2008–2011) conducted by the Korean Centers for Disease Control and Prevention. Prior to the survey, its content was explained to all subjects, and informed consent was obtained. The survey was conducted by trained interviewers. The researchers who created the questionnaire trained the interviewers to obtain appropriate information and to become familiarized with the matters that required attention for the interview by explaining the questionnaires and role-playing for four days.

Those who responded to both the examination survey and the health survey among adults aged 65 years or older who underwent whole-body dual-energy X-ray absorptiometry (DXA) were included in this study. Among the 37,753 individuals who participated in the KNHANES, 31,383 individuals who were <65 years old, 2682 individuals whose sarcopenia status and BMD were not measured, and 611 non-participants in the health survey because factors such as smoking, drinking status, and exercise frequency were excluded. Finally, 3077 individuals were selected (Figure 1).

### 2.2. Measurements of Variables

#### 2.2.1. Covariates

The physical examinations performed in this study included height, weight, diastolic and systolic blood pressure (DBP and SBP, respectively), waist circumference (WC), triglyceride level, fasting blood glucose level, and vitamin D measurements. BP was measured using a mercury sphygmomanometer in a seated position after a 10-min rest period. Two measurements were performed in all participants at 5-min intervals. An average of two measurements was used for the data analyses. Measurements of the vitamin D concentration, 25(OH)D3 (ng/mL), was performed using a 25-hydroxyvitamin D 125 I RIA Kit (DiaSorin, Stillwater, MN, USA). WC was measured at the midpoint between the bottom of the rib cage and the top of the lateral border of the iliac crest with full expiration. Blood samples were collected from participants in the morning after overnight fasting and were analyzed at a national central laboratory. Body mass index (BMI) was calculated by dividing the weight (kg) by height in meters squared (m^2^)]. Smoking status was categorized as never smokers, ex-smokers, and current smokers, while drinking status was categorized as current users and non-users.

The frequency of resistance exercise was assessed according to the participants’ answers to the question “How many times do you perform resistance exercise (push-ups, sit-ups, and lifting dumbbells or barbells) a week?” The short version of the International Physical Activity Questionnaire in Korea [39], which measures the health-related physical activity in populations, was used to measure the participants’ current walking. The number of days the participant walked ≥ 10 min at a time in the previous week was determined. Walking was measured based on the total walking time in a week (TWT) and calculated as follows: TWT = walking days (days/week) × walking minutes (minutes/day).

To assess the nutrient intake, all participants were instructed to maintain their usual dietary habits before the assessment of dietary intake. Daily food intake was measured using the 24-h recall method, while the daily nutrient intake was calculated using the Can-Pro 2.0, a nutrient intake assessment software developed by the Korean Nutrition Society (Seoul, Korea). The daily intake of proteins and calcium was also assessed.

#### 2.2.2. Measurement of Sarcopenia and Bone Mineral Density

Bone mineral content and body composition were measured by licensed technicians using DXA (Discovery QDR 4500 W, Hologic Inc., Belford, MA, USA). The participants fasted prior to the assessment and were placed in the supine position during the assessment. All non-fat and non-bone tissues were assumed to be skeletal muscles.

Appendicular skeletal muscle mass (ASM) was calculated as the sum of skeletal muscle mass in both arms and legs and was measured by DXA. The participants’ skeletal muscle mass index (SMI) was calculated as ASM (kg) divided by height in meters squared (m^2^). Sarcopenia was defined as SMI values of <7.0 kg/m^2^ for men and <5.4 kg/m^2^ for women, as recommended by the Asian Working Group for Sarcopenia (AWGS) [40].

BMD was measured at the lumbar spine, whole femur, and femoral neck using DXA. According to the World Health Organization criteria, the diagnosis of osteoporosis, osteopenia, and normal bone mass was determined based on the following T-scores: ≤−2.5, −2.5 to 1.0, and >−1.0, respectively, in the femur, femoral neck, or lumbar area [41].

### 2.3. Data Analysis

The data were analyzed using SPSS 27.0 Windows version (IBM, Armonk, NY, USA). This study was conducted as a cross-sectional study. As KNHANES is a sample survey rather than a census, all data analyses were performed using a weighted complex sample design. Therefore, the responses were weighted by multistage, complex, probability sampling. Continuous variables are presented as mean and standard deviation, and categorical variables as frequency and percentage (%). The Rao-Scott χ^2^ test or Student’s *t*-test or ANOVA was used to evaluate the differences in demographic and clinical characteristics by sex according to sarcopenia and osteoporosis. Complex sampling design multivariate logistic regression analysis adjusted for covariates was conducted to calculate ORs and 95% CIs and assess the association between sarcopenia, osteopenia, and osteoporosis. A *p*-value of <0.05 was considered significant.

## 3. Results

Table 1 shows the characteristics of the participants according to sex. The prevalence rates of sarcopenia were 41.8% in men and 38.4% in women, with no sex differences. The prevalence of osteopenia was significantly higher in men (53.1% vs. 37.0%), while the prevalence of osteoporosis was higher in women (12.6% vs. 59.9%). In addition, significant differences were observed in all variables except DBP, triglyceride level, fasting blood glucose level, and aerobic exercise.

Table 2 shows the characteristics of men and women affected by sarcopenia. For both men and women, participants with sarcopenia were older and had lower height, weight, WC, and BMI. In addition, both protein intake and calcium intake were significantly lower in participants with sarcopenia, while the prevalence of osteoporosis was high (6.2% vs. 21.6% in men and 54.9% vs. 67.9% in women). No difference was found in the prevalence of sarcopenia according to smoking status in men, but the current smoking rate was higher in women with sarcopenia. Moreover, the percentage of men with sarcopenia who did not exercise to improve muscle strength was high (69.5% vs. 78.4%), and the prevalence of aerobic exercise in this group was also high (58.00 vs. 72.41 in TWT).

Table 3 summarizes the characteristics of the osteoporosis group and osteopenia group stratified by sex. As age increased, the bone disorders of men and women became more severe. There were significant differences in the height, weight, BMI, and WC levels in the bone disorder (osteopenia, osteoporosis) group compared to the normal group.

The percentage of participants who did not exercise at all was significantly higher in the osteoporosis group than in the normal group (65.6% vs. 81.6% in men; 90.6% vs. 95.4% in women), and the difference in aerobic exercise was only found in women (82.61 vs. 59.89 in TWT). Protein intake was significantly lower in men and women than in women with osteoporosis, while calcium intake was lower only in men. Triglyceride and fasting blood glucose levels were also significantly lower in men with osteoporosis. The prevalence of sarcopenia was higher in osteopenia and osteoporosis than in normal individuals (22.0% vs. 47.5% vs. 71.4% in men and 15.3% vs. 32.0% vs. 43.6% in women).

Table 4 shows the association between sarcopenia and bone disorders, such as osteoporosis and osteopenia by sex. In model 1, which was not adjusted for any variables, the association between sarcopenia and bone disorders showed odds ratios of 3.211 (95% CI: 2.363–4.365) for osteopenia and 8.869 (95% CI: 5.517–14.259) for osteoporosis in men, and 2.609 (95% CI: 1.273–5.350) for osteopenia and 4.279 (95% CI: 2.092–8.755) for osteoporosis in women. On the contrary, after adjusting for all the covariates (model 4), the association between sarcopenia and bone disorders was significantly different in men (2.068, 95% CI: 1.462–2.924; 3.247, 95% CI: 1.953–5.399) but not in women. However, the sex-undifferentiated results showed odds ratios of 2.051 (95% CI: 1.498–2.808) for osteopenia and 2.258 (95% CI: 1.584–3.218) for osteoporosis.

## 4. Discussion

The main findings of this study were that sarcopenia is independently related to osteopenia and osteoporosis in older Korean adults, after adjusting for various confounding covariates. This association was particularly strong among men.

The prevalence rates of sarcopenia in Korea were 8.7% in men and 11.2% in women in those >20 years old [42], and 18.5% in men and 14.6% in women in those >40 years old [43]. In this study, 41.8% and 38.4% of men and women were >65 years old, respectively. As such, the rate of sarcopenia increases with age, and the rate of sarcopenia in men increases rapidly as the age increases compared with that in women [42,43].

The number of days of resistance exercise in older men with sarcopenia was significantly lower than normal, while aerobic exercise was more prevalent in those with sarcopenia (Table 2). These findings support that of a previous study, which showed that resistance exercise is more beneficial for improving sarcopenia than aerobic exercise [44]. Physical activity is effective in improving muscle strength and balance of the upper and lower extremities in patients with sarcopenia [45]. In addition, resistance exercise increases muscle strength and power by stimulating muscle hypertrophy and improving physical performance [46,47]. For these reasons, there was a significant difference in resistance exercise according to the prevalence of sarcopenia in this study.

No difference was observed in the prevalence of sarcopenia according to the type of exercise in women, whereas a difference was found in the rate of current smoking between men and women (Table 2). In a previous study, the current smoking rate was significantly higher in older women with sarcopenia [48], which is consistent with the results of this study. However, in this previous study, women who performed regular physical activity were about 50% more likely to develop sarcopenia, which contrasts with the results of this study [48]. This is thought to be a difference because resistance exercise and aerobic exercise were defined according to the frequency and time of exercise in this study, but the previous study simply classified physical activity as the performance of physical activity for three days or more.

Protein and calcium intakes were significantly lower in participants with sarcopenia, regardless of sex (Table 2). Protein, amino acids, and antioxidant intakes play important roles in preventing and alleviating sarcopenia [49]. In addition, protein intake, especially after performing a resistance training exercise, has a significantly positive effect on the treatment outcomes of sarcopenia [50]. Also, minerals such as calcium and magnesium are known to be important nutrients that help prevent and treat sarcopenia [51,52]. This evidence is also supported by the results of the comparison of calcium intakes according to sarcopenia status in this study [49].

In this study, the prevalence rates of osteopenia were 53.1% in men and 37.0% in women, while the prevalence rates of osteoporosis were 12.6% in men and 59.9% in women (Table 1). In previous studies, the prevalence rates of osteopenia were 48.6% and 48.0% in men and women, while the prevalence rates of osteoporosis were 8.8% and 39.1% in men and women, respectively [53]. The reason for this difference in prevalence seems to be because the participants selected were >65 years old, while the participants included in previous studies were >50 years old.

Table 3 shows the characteristics of men and women participants according to bone disorders such as osteopenia and osteoporosis. In osteoporosis, the rate of no resistance exercise was the highest (81.6% in men, 95.4% in women). On the contrary, aerobic exercise was significantly lower in women with osteopenia and osteoporosis than in women. Resistance exercise has a beneficial effect on the physical function and the performance of activities of daily living in patients with osteopenia and osteoporosis [54]. In addition, BMD has a positive correlation with exercise [55,56], meanwhile, muscle strength exercise is considered to be a powerful stimulus to improve and maintain bone mass compared with low-intensity aerobic exercise such as walking [56,57]. In this study, only resistance exercise differed substantially according to the bone disorder in men. On the other hand, the lower BMD, the lower the resistance and aerobic exercise in women; and for these reasons, the incidence of osteoporosis is significantly higher in women than in men. In addition, more osteoporosis may occur in postmenopausal women due to a decrease in estrogen secretion [58].

The mechanisms underlying the association between sarcopenia and bone disorders, such as osteopenia and osteoporosis, can be explained in several ways. First, it can be explained based on the “mechanostat theory,” in which mechanical stimulation of the muscles affects the formation and recovery of the bones [59]. In other words, the muscles have long been regarded as a major source of anabolism in bone tissues [18]. However, the precise mechanisms for controlling bone and skeletal muscle mass remain unclear, and little is known about its potential mechanisms [60]. Second, the muscles secrete insulin-like growth factors-1 (IGF-1) and fibroblast growth factor-2 that promote bone formation [61,62], while myostatin and pro-inflammatory cytokines activate bone destruction [63]. This may be due to an imbalance in several factors capable of maintaining the bones and muscles. Third, vitamin D plays an important role not only in bone metabolism but also in muscle metabolism. Vitamin D binds to muscle cells to promote protein synthesis and stimulate the movement of calcium through the cell membranes [64,65,66]. However, in this study, no significant difference was observed in the vitamin D levels according to the sarcopenia status and bone disorders. This is thought to be the result of an overall vitamin D deficiency in older Korean adults. Vitamin D deficiency is defined as a vitamin D level of less than 30 ng/mL (75 nmol/L) in older adults [67]. In this study, the average vitamin D levels by sex were 21.49 ± 0.33 ng/mL in men and 18.61 ± 0.23 ng/mL in women, indicating a deficiency. Therefore, it affected the overall decrease in skeletal muscle mass and the change in bone mass in older participants.

After adjusting for many confounding variables affecting the prevalence of sarcopenia and bone disorder, the association between sarcopenia and bone disorders was only observed in older men. Aging causes bone and muscle mass loss [68], which are affected by sex differences [69]. Changes in bone and muscle tend to increase the muscle mass due to the increased levels of growth hormones such as testosterone, somatotropin, and IGF-1 in men, which results in a rapid increase in bone mass [70,71]. In women, only estrogen levels were found to be related to bone density [72].

Recent randomized controlled trials show that resistance exercise and sufficient intake of nutrients such as protein and vitamin D improve the muscle strength and physical performance of the older frail [73]. In addition, high protein intake has a positive effect on bone density and causes a decrease in bone loss and an increase in muscle strength [74]. Also, proteins interact with nutrients such as vitamin D and calcium to regulate bone and muscle metabolism [75]. A previous study found that proper correction of vitamin D and calcium deficiency in the elderly with sarcopenia and osteoporosis at the same time increases BMD and reduces the risk of fracture [75,76]. Moreover, when more protein and adequate calcium intake were combined, older adults showed a higher BMD [77]. As shown in Table 3, protein intake differed significantly according to bone disorders in both men and women, but calcium intake was significantly lower only in men with bone disorders. Due to insufficient nutritional intake and hormonal effects, the association between sarcopenia and bone disorder was significant only in older men.

Taken together, the results of this study revealed that sarcopenia is significantly related to osteopenia and osteoporosis in the elderly. In addition, therapeutic management is required through the proper provision of physical activity, especially resistance exercise, and nutrients such as hyper protein, vitamin D, and calcium is to improve falls, fractures, and weaknesses in these patients.

These KNHANES data have the following strengths: the response rates and accuracy as measured by professional medical staff were high. Despite some meaningful findings in this study, there are some limitations that must be taken into consideration while evaluating the results of this study. First, the individuals who participated in the KNHANES had a relatively mild level of disease; therefore, the non-participation of a small number of severe sarcopenia or osteoporosis patients may affect the analysis of the results. However, since these data were obtained from the national population, potential disturbing factors will not have a significant impact on the results. Second, a new guideline for the 2019 AWGS suggested measuring muscle strength or physical performance as well as muscle mass when diagnosing sarcopenia. However, this was not measured in the KNHANES. Third, although this study found the association between sarcopenia and bone disorder by adjusting various covariates, it was a cross-sectional study that simultaneously measured sarcopenia and bone disorder factors. Therefore, the temporal concern could not be determined, which made it impossible to accurately identify the order of fundamental causes between the two diseases. Therefore, it will be valuable to uncover the mechanism that can clarify the causal relationship between the two through future longitudinal studies.

## 5. Conclusions

This study was conducted to determine the association between sarcopenia and bone disorders in older Korean adults. Sarcopenia is significantly related to osteopenia and osteoporosis, especially in men, when stratified by sex. To improve falls, fractures, weaknesses, and the incidence of these patients, therapeutic management is necessary through the proper provision of physical activity, particularly resistance training, and nutrients such as high protein, vitamin D, and calcium.

## Figures and Tables

**Figure 1 jcm-11-00129-f001:**
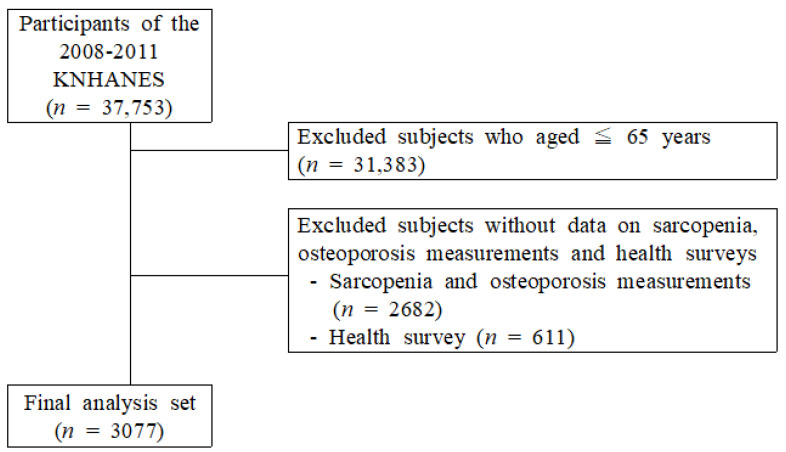
Selection of participants from the Korea National Health and Nutrition Examination Survey 2008–2011.

**Table 1 jcm-11-00129-t001:** General characteristics of participants by sex.

Variables	Total (*n* = 3077)	Men (*n* = 1376)	Women (*n* = 1701)	*p*
Age (y)	71.82 ± 0.12	71.40 ± 0.15	72.14 ± 0.15	<0.001
Height (cm)	157.07 ± 0.20	165.01 ± 0.19	150.96 ± 0.17	<0.001
Weight (kg)	58.64 ± 0.25	63.11 ± 0.34	55.20 ± 0.27	<0.001
SBP (mmHg)	131.74 ± 0.47	130.22 ± 0.65	132.91 ± 0.59	<0.001
DBP (mmHg)	76.93 ± 0.26	77.09 ± 0.35	76.81 ± 0.34	0.534
WC (cm)	84.00 ± 0.24	84.67 ± 0.32	83.48 ± 0.30	0.003
Triglyceride	146.97 ± 2.01	142.72 ± 3.13	150.23 ± 2.71	0.076
Fasting glucose (mg/dL)	104.41 ± 0.56	104.50 ± 0.75	104.34 ± 0.81	0.887
BMI (kg/m^2^)	23.72 ± 0.08	23.13 ± 0.11	24.17 ± 0.10	<0.001
<18.5 (underweight), *n* (%)	135 (4.3)	77 (5.5)	58 (3.4)	
<25 (normoweight), *n* (%)	1960 (63.5)	962 (70.0)	998 (58.5)	
≥25 (overweight), *n* (%)	982 (32.2)	337 (24.5)	645 (38.2)	
Smoking status (%) (current-/ex-/nonsmoker)	30.2/13.6/56.2	57.4/28.7/14.0	9.3/1.9/88.8	<0.001
Drinking status (%) (current-/nondrinking)	49.3/50.7	68.8/31.2	34.3/65.7	<0.001
Resistance exercise				<0.001
Never	2596 (84.2)	1022 (73.2)	1574 (92.7)	
1–3 days/wk.	229 (7.9)	162 (12.7)	67 (4.2)	
≥4 days/wk.	252 (7.9)	192 (14.1)	60 (3.1)	
Aerobic exercise (TWT)	65.22 ± 1.92	64.03 ± 2.76	66.14 ± 2.61	0.573
25(OH)D (ng/mL)	19.87 ± 0.26	21.49 ± 0.33	18.61 ± 0.23	<0.001
Protein intake (g)	53.15 ± 0.72	64.29 ± 1.09	44.57 ± 0.65	<0.001
Calcium intake (mg)	419.86 ± 11.53	484.19 ± 11.87	370.35 ± 15.17	<0.001
Sarcopenia, *n* (%)	1230 (39.9)	585 (41.8)	645 (38.4)	0.133
ASM (kg)	16.11 ± 0.08	19.53 ± 0.10	13.48 ± 0.06	<0.001
SMI (kg/m^2^)	6.45 ± 0.02	7.16 ± 0.03	5.90 ± 0.02	<0.001
Osteopenia (%)	1,402 (44.0)	750 (53.1)	652 (37.0)	<0.001
Osteoporosis (%)	1,156 (39.9)	166 (12.6)	990 (59.9)	<0.001
Total femur BMD (g/cm^2^)	0.78 ± 0.00	0.88 ± 0.00	0.71 ± 0.00	<0.001
Femoral neck BMD (g/cm^2^)	0.62 ± 0.00	0.70 ± 0.00	0.56 ± 0.00	<0.001
Total body BMD (g/cm^2^)	1.04 ± 0.00	1.14 ± 0.00	0.96 ± 0.00	<0.001
Total femur T-score	−0.91 ± 0.02	−0.44 ± 0.03	−1.27 ± 0.03	<0.001
Femoral neck T-score	−1.80 ± 0.02	−1.16 ± 0.03	−2.29 ± 0.03	<0.001

Data were presented as means ± SD or number (%). differences SBP, systolic blood pressure; DBP, diastolic blood pressure; WC, waist circumference; BMI, body mass index; TWT, total walking time in a week; BMD, bone mineral density; ASM, appendicular skeletal muscle mass; SMI, skeletal muscle mass index.

**Table 2 jcm-11-00129-t002:** Characteristics of participants stratified by sex according to sarcopenia.

Variables	Men (*n* = 1376)			Women (*n* = 1701)		
	Non-Sarcopenia (*n* = 791)	Sarcopenia (*n* = 585)	*p*	Non-Sarcopenia (*n* = 1056)	Sarcopenia (*n* = 645)	*p*
Age (y)	70.54 ± 0.20	72.60 ± 0.23	<0.001	71.57 ± 0.18	73.05 ± 0.24	<0.001
Height (cm)	165.54 ± 0.25	164.27 ± 0.28	<0.001	151.29 ± 0.19	150.44 ± 0.32	0.024
Weight (kg)	67.40 ± 0.36	57.13 ± 0.38	<0.001	58.27 ± 0.31	50.29 ± 0.36	<0.001
Systolic BP (mmHg)	130.34 ± 0.77	130.06 ± 1.03	0.815	133.13 ± 0.72	132.55 ± 0.94	0.611
Diastolic BP (mmHg)	77.80 ± 0.43	76.11 ± 0.57	0.015	77.48 ± 0.43	75.73 ± 0.52	0.009
Waist circumference (cm)	88.03 ± 0.34	80.00 ± 0.44	<0.001	86.15 ± 0.34	79.21 ± 0.40	<0.001
Triglyceride	145.42 ± 4.00	138.97 ± 4.69	0.280	151.29 ± 3.67	148.53 ± 3.65	0.586
Fasting glucose (mg/dL)	104.51 ± 0.98	104.49 ± 1.16	0.990	104.93 ± 0.91	103.40 ± 1.45	0.363
BMI (kg/m^2^), mean (SD)	24.56 ± 0.11	21.14 ± 0.11	<0.001	25.40 ± 0.11	22.20 ± 0.13	<0.001
<18.5 (underweight), *n* (%)	1 (0.2)	76 (12.9)		11 (1.0)	47 (7.1)	
<25 (normoweight), *n* (%)	478 (61.2)	484 (82.2)		486 (45.0)	512 (80.0)	
≥25 (overweight), *n* (%)	312 (38.6)	25 (4.8)		559 (54.0)	86 (12.9)	
Smoking status (%) (current-/ex-/nonsmoker)	56.8/28.8/14.4	58.2/28.4/13.4	0.867	7.3/2.1/90.5	12.4/1.6/85.9	0.019
Drinking status (%) (current-/nondrinking)	71.5/28.5	65.0/35.0	0.031	35.0/65.0	33.1/66.9	0.498
Resistance exercise			0.006			0.924
Never	557 (69.5)	465 (78.4)		978 (92.7)	596 (92.6)	
1–3 days/wk.	108 (15.1)	54 (9.3)		42 (4.1)	25 (4.4)	
≥4 days/wk.	126 (15.4)	66 (12.3)		36 (3.2)	24 (3.0)	
Aerobic exercise (TWT)	58.00 ± 3.60	72.41 ± 4.41	0.013	63.29 ± 3.29	70.73 ± 40.8	0.151
25(OH)D (ng/mL)	21.69 ± 0.42	21.22 ± 0.41	0.366	18.81 ± 0.34	18.29 ± 0.40	0.259
Protein intake (g)	67.93 ± 1.31	59.22 ± 1.56	<0.001	46.49 ± 0.84	41.50 ± 1.01	<0.001
Calcium intake (mg)	521.72 ± 15.53	431.92 ± 15.46	<0.001	399.94 ± 23.65	322.96 ± 10.56	0.003
ASM (kg)	21.10 ± 0.09	17.53 ± 0.09	<0.001	14.43 ± 0.05	11.95 ± 0.07	<0.001
SMI (kg/m^2^)	7.69 ± 0.02	6.42 ± 0.02	<0.001	6.29 ± 0.02	5.27 ± 0.02	<0.001
Osteopenia /Osteoporosis (%)	47.9/6.2	60.4/21.6	<0.001	40.8/54.9	30.8/67.9	<0.001
Total femur BMD (g/cm^2^)	0.92 ± 0.00	0.83 ± 0.00	<0.001	0.73 ± 0.00	0.67 ± 0.00	<0.001
Femoral neck BMD (g/cm^2^)	0.74 ± 0.00	0.66 ± 0.00	<0.001	0.57 ± 0.00	0.53 ± 0.00	<0.001
Total body BMD (g/cm^2^)	1.16 ± 0.00	1.12 ± 0.01	<0.001	0.96 ± 0.00	0.94 ± 0.00	0.009
Total femur T-score	−0.16 ± 0.04	−0.83 ± 0.04	<0.001	−1.09 ± 0.03	−1.55 ± 0.04	<0.001
Femoral neck T-score	−0.89 ± 0.04	−1.53 ± 0.04	<0.001	−2.16 ± 0.03	−2.51 ± 0.04	<0.001

Data were presented as means ± SD or number (%). differences SBP, systolic blood pressure; DBP, diastolic blood pressure; WC, waist circumference; BMI, body mass index; TWT, total walking time in a week; BMD, bone mineral density; ASM, appendicular skeletal muscle mass; SMI, skeletal muscle mass index.

**Table 3 jcm-11-00129-t003:** Characteristics of participants stratified by sex according to bone disorders.

Types of Bone Disorder	Men (*n* = 1376)				Women (*n* = 1701)			
	Normal (*n* = 460)	Osteopenia (*n* = 750)	Osteoporosis (*n* = 166)	*p*	Normal (*n* = 59)	Osteopenia (*n* = 652)	Osteoporosis (*n* = 990)	*p*
Age (y)	70.27 ± 0.22 ^a^	71.75 ± 0.20 ^b^	73.02 ± 0.44 ^c^	<0.001	68.67 ± 0.48 ^a^	70.73 ± 0.20 ^b^	73.19 ± 0.19 ^c^	<0.001
Height (cm)	166.51 ± 0.32 ^a^	164.79 ± 0.23 ^b^	161.84 ± 0.58 ^c^	<0.001	153.58 ± 1.17 ^a^	152.58 ± 0.24 ^b^	149.83 ± 0.22 ^c^	<0.001
Weight (kg)	68.02 ± 0.48 ^a^	61.83 ± 0.37 ^b^	55.13 ± 0.83 ^c^	<0.001	63.81 ± 1.29 ^a^	58.84 ± 0.36 ^b^	52.50 ± 0.32 ^c^	<0.001
Systolic BP (mmHg)	131.58 ± 1.10	129.07 ± 0.79	131.38 ± 1.69	0.392	135.28 ± 2.42	132.47 ± 0.91	133.05 ± 0.75	0.753
Diastolic BP (mmHg)	78.11 ± 0.56	76.57 ± 0.47	76.51 ± 0.94	0.280	80.50 ± 1.53 ^a^	77.28 ± 0.47 ^a b^	76.33 ± 0.43 ^b^	0.016
Waist circumference (cm)	88.22 ± 0.44 ^a^	83.85 ± 0.40 ^b^	78.52 ± 0.80 ^c^	<0.001	89.39 ± 1.11 ^a^	86.13 ± 0.42 ^b^	81.54 ± 0.37 ^c^	<0.001
Triglyceride	153.66 ± 5.81 ^a^	138.12 ± 4.01 ^b c^	132.39 ± 6.76 ^c^	0.043	134.06 ± 12.02	149.25 ± 4.27	151.68 ± 3.56	0.310
Fasting glucose (mg/dL)	108.84 ± 1.42 ^a^	102.69 ± 1.00 ^b^	100.35 ± 1.57 ^b^	<0.001	106.91 ± 3.27	105.27 ± 1.18	103.63 ± 1.08	0.585
BMI (kg/m^2^), mean (SD)	24.51 ± 0.16 ^a^	22.74 ± 0.12 ^b^	21.01 ± 0.56 ^c^	<0.001	26.98 ± 0.35 ^a^	25.26 ± 0.14 ^b^	23.35 ± 0.13 ^c^	<0.001
<18.5 (underweight), *n* (%)	2 (0.8)	45 (5.5)	30 (18.6)		0 (0)	7 (0.7)	51 (5.2)	
<25 (normoweight), *n* (%)	276 (59.6)	565 (75.9)	121 (73.4)		17 (22.9)	324 (50.2)	657 (65.4)	
≥25 (overweight), *n* (%)	182 (39.6)	140 (18.6)	15 (8.0)		42 (77.1)	321 (49.1)	282 (29.4)	
Smoking status (%) (current-/ex-/nonsmoker)	52.2/30.7/17.0	59.9/27.6/12.5	60.7/27.4/12.0	0.117	2.7/2.3/95.0 ^a^	4.8/1.7/93.5 ^a^	12.4/2.1/85.6 ^b^	0.002
Drinking status (%) (current-/nondrinking)	71.7/28.3	69.1/30.9	59.7/40.3	0.042	44.5/55.5	69.1/30.9	59.7/40.3	0.001
Resistance exercise				0.003				<0.001
Never	312 (65.6)	573 (76.2)	137 (81.6)		52 (90.6)	584 (88.4)	938 (95.4)	
1–3 days/wk.	60 (16.2)	92 (12.1)	10 (5.7)		3 (4.0)	34 (6.4)	30 (2.8)	
≥4 days/wk.	88 (18.2)	85 (11.8)	19 (12.7)		4 (5.3)	34 (5.2)	22 (1.8)	
Aerobic exercise (TWT)	62.67 ± 4.70	63.81 ± 3.44	68.62 ± 8.40	1.000	82.61 ± 19.29 ^a^	74.87 ± 3.92 ^b^	59.89 ± 3.55 ^c^	0.009
25(OH)D (ng/mL)	21.78 ± 0.42	21.32 ± 0.40	21.45 ± 0.74	1.000	19.14 ± 1.24	19.25 ± 0.41	18.19 ± 0.36	0.086
Protein intake (g)	69.28 ± 1.59 ^a^	62.56 ± 1.43 ^b^	58.07 ± 2.77 ^b^	<0.001	52.31 ± 3.26 ^a^	46.70 ± 1.04 ^b^	42.85 ± 0.75 ^b^	0.002
Calcium intake (mg)	538.39 ± 17.52 ^a^	469.71 ± 16.70 ^b^	397.91 ± 27.20 ^c^	<0.001	445.25 ± 50.88	391.52 ± 14.39	353.37 ± 23.34	0.193
Sarcopenia, *n* (%)	112 (22.0)	357 (47.5)	116 (71.4)	<0.001	14 (15.3)	208 (32.0)	423 (43.6)	<0.001
ASM (kg)	20.84 ± 0.15 ^a^	19.16 ± 0.12 ^b^	17.54 ± 0.22 ^c^	<0.001	15.05 ± 0.28 ^a^	14.05 ± 0.09 ^b^	13.04 ± 0.07 ^c^	<0.001
SMI (kg/m^2^)	7.50 ± 0.04 ^a^	7.05 ± 0.04 ^b^	6.68 ± 0.06 ^c^	<0.001	6.37 ± 0.11 ^a^	6.03 ± 0.03 ^b^	5.80 ± 0.03 ^c^	<0.001
Total femur BMD (g/cm^2^)	0.82 ± 0.00 ^a^	0.84 ± 0.00 ^b^	0.72 ± 0.00 ^c^	<0.001	0.92 ± 0.01 ^a^	0.77 ± 0.00 ^b^	0.65 ± 0.00 ^c^	<0.001
Femoral neck BMD (g/cm^2^)	0.82 ± 0.00 ^a^	0.66 ± 0.00 ^b^	0.55 ± 0.00 ^c^	<0.001	0.77 ± 0.01 ^a^	0.62 ± 0.00 ^b^	0.51 ± 0.00 ^c^	<0.001
Total body BMD (g/cm^2^)	1.24 ± 0.00 ^a^	1.11 ± 0.00 ^b^	1.02 ± 0.00 ^c^	<0.001	1.12 ± 0.02 ^a^	1.01 ± 0.00 ^b^	0.91 ± 0.00 ^c^	<0.001
Total femur T-score	0.45 ± 0.03 ^a^	−0.73 ± 0.02 ^b^	−1.64 ± 0.06 ^c^	<0.001	0.58 ± 0.10 ^a^	−0.70 ± 0.03 ^b^	−1.71 ± 0.03 ^c^	<0.001
Femoral neck T-score	−0.21 ± 0.03 ^a^	−1.48 ± 0.02 ^b^	−2.36 ± 0.06 ^c^	<0.001	−0.32 ± 0.09 ^a^	−1.75 ± 0.02 ^b^	−2.73 ± 0.03 ^c^	<0.001

Data were presented as means ± SD or number (%). a, b, c, the same letters indicate the non-significant difference between groups based on Bonferroni multiple comparison test. SBP, systolic blood pressure; DBP, diastolic blood pressure; WC, waist circumference; BMI, body mass index; TWT, total walking time in a week; BMD, bone mineral density; ASM, appendicular skeletal muscle mass; SMI, skeletal muscle mass index.

**Table 4 jcm-11-00129-t004:** Odds ratios for osteopenia and osteoporosis stratified by sex according to sarcopenia status.

		Odds Ratio (95% CI)		
		Total	Men	Women
Model 1	normal	1	1	1
	osteopenia	2.482 (1.894–3.253) ***	3.211 (2.363–4.365) ***	2.609 (1.273–5.350) ***
	osteoporosis	3.343 (2.533–4.412) ***	8.869 (5.517–14.259) ***	4.279 (2.092–8.755) ***
Model 2	normal	1	1	1
	osteopenia	3.038 (2.282–4.043) ***	2.935 (2.147–4.011) ***	2.368 (1.164–4.815) *
	osteoporosis	5.003 (3.591–6.970) ***	7.554 (4.617–12.362) ***	3.421 (1.677–6.978) ***
Model 3	normal	1	1	1
	osteopenia	2.087 (1.523–2.860) ***	2.108 (1.488–2.987) ***	1.308 (0.616–2.776)
	osteoporosis	2.264 (1.587–3.229) ***	3.401 (2.030–5.698) ***	1.252 (0.598–2.621)
Model 4	normal	1	1	1
	osteopenia	2.051 (1.498–2.808) ***	2.068 (1.462–2.924) **	1.303 (0.621–2.735)
	osteoporosis	2.258 (1.584–3.218) ***	3.247 (1.953–5.399) **	1.275 (0.615–2.646)

Model 1: crude; Model 2: adjusted for age and sex; Model 3: adjusted for variables in model 2 + BMI, smoking status, drinking status, waist circumference, triglyceride, SBP, DBP, and fasting glucose; Model 4: adjusted for variables in model 3 + resistance exercise, aerobic exercise, 25(OH)D, protein intake, and calcium intake; * *p* < 0.05, ** *p* < 0.01, *** *p* < 0.001.

## Data Availability

All data were anonymized and can be downloaded from the website at https://knhanes.kdca.go.kr/knhanes (accessed on 9 November 2021).

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
