# Peer review of "Association of Sarcopenia with Osteopenia and Osteoporosis in Community-Dwelling Older Korean Adults: A Cross-Sectional Study"

_jcm, 2021, doi:10.3390/jcm11010129_

Round 1

Reviewer 1 Report

Thanks to the authors for their work.

This is an interesting paper on a very actual topic, which investigate the possible association of sarcopenia with osteopenia and osteoporosis, in a sample of community-dwelling Korean adults.

I find that this study was conducted with a rigorous and easily replicable methodology.

As a unique suggestion, I would like to ask if the distribution of the variables has been assessed before the analysis. If it has been done I would specify it in the text, otherwise I suggest to verify that the tests used are appropriate, based on the distribution of variables

Author Response

Response to Reviewer 1

Thank you for your review of our paper. We have answered each of your points below.

  1. As a unique suggestion, I would like to ask if the distribution of the variables has been assessed before the analysis. If it has been done I would specify it in the text, otherwise I suggest to verify that the tests used are appropriate, based on the distribution of variables.

Author response: Thank you for your feedback. KNHANES data is the official national disclosure data conducted annually. KNHANES is a survey research program conducted by the Korean Centers for Diseases Control and Prevention to assess the health and nutritional status of adults and children (0~80 years old) in the Korea. The survey combines interviews, physical examinations, and laboratory tests. Therefore, all examination data used in this survey were based on data with proven validity.

Reviewer 2 Report

Dear Authors,

The topic is very interesting considering the gap of knowledge regarding the relationship between sarcopenia and osteoporosis in men, or the variables that could affect the two diseases, such as lifestyle, exercise status, and nutrition factors in both sexes. Moreover, the results are intriguing considering the large sample size (n=3077).

However, I have doubts about the methodological implant of the study and some critical issues should be addressed.

Major revisions

TITLE. Please, clarify the study’s design in the title. You should also highlight the number of patients included.

INTRODUCTION. The introduction section should be improved focusing on the disabling sequelae of osteoporosis, sarcopenia, and fragility fractures and underlining the challenges in the therapeutic management.

According to this, you should cite the following references:

- de Sire A et al. Optimization of transdisciplinary management of elderly with femur proximal extremity fracture: A patient-tailored plan from orthopaedics to rehabilitation. World J Orthop. 2021 Jul 18;12(7):456-466. doi: 10.5312/wjo.v12.i7.456.

- Lorentzon M et al.. Osteoporosis and fractures in women: the burden of disease. Climacteric. 2021 Jul 28:1-7. doi: 10.1080/13697137.2021.1951206.

- Lombardi F et al. Underprescription of medications in older adults: causes, consequences and solutions-a narrative review. Eur Geriatr Med. 2021 Jun;12(3):453- 462. doi: 10.1007/s41999-021-00471-x.

METHODS. Please, study design should be clarified in this section.

METHODS. Please, provide information about subjects’ informed consent.

METHODS. The Methods of sample selection used by the Korea National Health and Nutrition Examination Survey (KNHANES) should be better clarified.

METHODS. Survey realization methods should be clearly presented.

METHODS. Please, report the results of patient selection process in the “Results” Section. Accordingly, Figure 1 should be reported in the “Results” section.

RESULTS. Patients assessed for eligibility and patients excluded should be clarified in the results section, characterizing at least the main cause of exclusions.

DISCUSSION. The discussion section should be improved, highlighting the role of physical activity and nutritional approach in counteracting functional consequences of sarcopenia and osteoporosis.

According to this, you should cite the following references:

  • Invernizzi M et al. Effects of essential amino acid supplementation and rehabilitation on functioning in hip fracture patients: a pilot randomized controlled trial. Aging clinical and experimental research, 2019, 31(10), 1517–1524. https://doi.org/10.1007/s40520-018-1090-y
  • Kirk B et al. Nutrients to mitigate osteosarcopenia: the role of protein, vitamin D and calcium. Current opinion in clinical nutrition and metabolic care. 2021, 24(1), 25–32. https://doi.org/10.1097/MCO.0000000000000711

Minor revisions

INTRODUCTION. Page 2, line 48. Please, correct the wrongfully starting of a new paragraph in the middle of a sentence.

INTRODUCTION and DISCUSSION. Some sentences are not supported by references. Please provide appropriate reference citations.

DISCUSSION. Page 10, line 275. Please move the sentence “Despite these limitations, these data have the following strengths: the response rates and accuracy as measured by professional medical staff were high.” at the end of the paragraph to allow consecution tempore of the points you mentioned as limitations.

TABLE 1, TABLE 2 & TABLE 3. ASM and SMI abbreviations should be clarified at the foot of the tables.

Author Response

Response to Reviewer 2

Thank you for your comments. We have answered each of your points below.

  1. Please, clarify the study’s design in the title. You should also highlight the number of patients included.

Author response: We accepted your opinion and revised it. We added the study’s design “A Cross-Sectional Study” in the title. However, putting the number of patients in the title seems to be a matter to be considered one more time. The final number of patients can be sufficiently confirmed in “Data source and sampling” of the text and Figure 1. If my response does not fit your opinion direction that needs to be corrected, we will try to fix it as much as we can.

  1. The introduction section should be improved focusing on the disabling sequelae of osteoporosis, sarcopenia, and fragility fractures and underlining the challenges in the therapeutic management.

Author response: The introduction part additionally describes the importance of therapeutic management. Also, introduction content was partially modified to lead smoothly. Please check the contents in line 54-63.

  1. Please, study design should be clarified in this section.

Author response: The research design was revised and described in the title and data analysis section of the paper.

  1. Please, provide information about subjects’ informed consent.

METHODS. Survey realization methods should be clearly presented.

Author response: We added subject’ informed consent in “2.1. Data source and sampling” section. In addition, the survey realizations were described in more detail.

  1. The Methods of sample selection used by the Korea National Health and Nutrition Examination Survey (KNHANES) should be better clarified.

METHODS. Please, report the results of patient selection process in the “Results” Section. Accordingly, Figure 1 should be reported in the “Results” section.

RESULTS. Patients assessed for eligibility and patients excluded should be clarified in the results section, characterizing at least the main cause of exclusions.

Author response: KNHANES data is the official national disclosure data conducted annually. KNHANES is a survey research program conducted by the Korean Centers for Diseases Control and Prevention to assess the health and nutritional status of adults and children (0~80 years old) in the Korea. The survey combines interviews, physical examinations, and laboratory tests.

Therefore, In the total KNHANES data sampled from the whole country, subjects with variables to be analyzed in this study were selected. The subjects necessary for this study were selected according to age, sarcopenia and osteoporosis variables measured, and whether they participated in the examination and survey. These contents were described in the “2.1. Data source and sampling” and “Figure 1” section, and necessary matters were additionally modified.

In most previous studies*, it was referenced that the subject selection method and diagram are described in the research method. For this reason, the patient selection part was written in the "Methods" part. If the part that needs to be revised in my paper does not fit your opinion direction, I will try to fix it as much as possible, so please let me know the details.

*

  • Huh, Da-An, Yun-Hee Choi, and Kyong Whan Moon. "The effects of earphone use and environmental lead exposure on hearing loss in the Korean population: data analysis of the Korea National Health and Nutrition Examination Survey (KNHANES), 2010–2013." PloS one 11.12 (2016): e0168718.
  • Kim, Yangho, and Byung-Kook Lee. "Association between urinary arsenic and diabetes mellitus in the Korean general population according to KNHANES 2008." Science of the Total Environment 409.19 (2011): 4054-4062.

Also, for understanding KNHANES data, please refer to the paper below.

  • Kweon, Sanghui, et al. "Data resource profile: the Korea national health and nutrition examination survey (KNHANES)." International journal of epidemiology1 (2014): 69-77.

  1. The discussion section should be improved, highlighting the role of physical activity and nutritional approach in counteracting functional consequences of sarcopenia and osteoporosis.

According to this, you should cite the following references:

  • Invernizzi M et al. Effects of essential amino acid supplementation and rehabilitation on functioning in hip fracture patients: a pilot randomized controlled trial. Aging clinical and experimental research, 2019, 31(10), 1517–1524. https://doi.org/10.1007/s40520-018-1090-y
  • Kirk B et al. Nutrients to mitigate osteosarcopenia: the role of protein, vitamin D and calcium. Current opinion in clinical nutrition and metabolic care. 2021, 24(1), 25–32. https://doi.org/10.1097/MCO.0000000000000711

Author response: We found out from your review that there was something we missed. Therefore, we revised the discussion section overall according to the reference you recommended. Please check again if there is anything wrong with this part (line 286-305).

  1. Page 2, line 48. Please, correct the wrongfully starting of a new paragraph in the middle of a sentence.

Author response: We accepted your opinion and revised it.

  1. INTRODUCTION and DISCUSSION. Some sentences are not supported by references. Please provide appropriate reference citations.

Author response: Thank you for your detail review. We revised it. (Ref. no. 42, 43, 48, 49)

  1. Page 10, line 275. Please move the sentence “Despite these limitations, these data have the following strengths: the response rates and accuracy as measured by professional medical staff were high.” at the end of the paragraph to allow consecution tempore of the points you mentioned as limitations.

Author response: Thank you for feedback. The sentence was moved to the beginning of the paragraph.

  1. TABLE 1, TABLE 2 & TABLE 3. ASM and SMI abbreviations should be clarified at the foot of the tables.

Author response: Thank you for your advice. We revised it.

Reviewer 3 Report

Lee and Shin report an association of sarcopenia with osteopenia and osteoporosis in a corean cohort.

Overall, the manuscript requires revisions by a native speaker in order to make the text readable.

Introduction

Please refer to the term "wasting" and "cachexia", and dissociate these from osteoporosis and sarcopenia.

The rationale for a better understanding of the covariate "sex" should be worked out more profoundly.

Materials and Methods

When was 25-hydroxyvitamin D 125 measured? Was it at the exact same time of day (just "morning" is mentioned)?

Is the "The short version of the International 91 Physical Activity Questionnaire in Korea" validated?

Sarcopenia should not just be defined upon SMI. At least, the EWGSOP2 criteria should be used.

Was normality tested before a chi-square or students t- test were used?

Results

Since the two groups are not homogenous, a comparison seems to be difficult to control for all these covariates.

But was protein intake and calcium intake lower due to sarcopenia, or were these two leading to sarcopenia?

"as age increased, the BMD levels of men and women decreased" - where is the correlation (r, p?)? How was the correlation calculated?

"Height, weight, BMI, and WC tended to decrease as the degree of bone disorder increased." see above: where is the correlation?

For which covariates was adjusted? All? Just significant differences from table 1? Please be specific.

Why show models 1-3? All covariates with significant differences in table 1 should be adjusted for.

Discussion

Again, a correlation is missing.

"Third, although this study may help provide more information about the 277 nature of the relationship, it was a cross-sectional study that simultaneously measured 278 sarcopenia and bone disorder factors." - I think this is not true. You may have showed some associations, but did not provide causal connections between factors. In order to do so, a prosepective randomized trial is needed.

Author Response

Response to Reviewer 3

Thank you for your feedback of our paper. Our answers to your points are as follows.

  1. Introduction
    Please refer to the term "wasting" and "cachexia” and dissociate these from osteoporosis and sarcopenia.

Author response: Thank you for your review. We revised the first paragraph in the introduction part based on the terms of “wasting” and “cachexia” you mentioned. If my response does not fit your opinion direction that needs to be corrected, we will try to fix it as much as we can. Please, check again this part. (line 27-32)

  1. Introduction
    The rationale for a better understanding of the covariate "sex" should be worked out more profoundly.

Author response: Thank you for your advice. We added a reason to classify this study by sex. Please, check line 66-71 in introduction part.

  1. Materials and Methods
    When was 25-hydroxyvitamin D 125 measured? Was it at the exact same time of day (just "morning" is mentioned)?

Author response: All examinations are usually open from 6 a.m. to 1 p.m. Also, based on adults, it takes a total of 1 hour and 30 minutes to 2 hours per person.
For understanding KNHANES data, please refer to the paper below.

  • Kweon, Sanghui, et al. "Data resource profile: the Korea national health and nutrition examination survey (KNHANES)." International journal of epidemiology1 (2014): 69-77.

Is the "The short version of the International 91 Physical Activity Questionnaire in Korea" validated?

Author response: Thank you for your feedback. KNHANES data is the official national disclosure data conducted annually. Therefore, all examination data used in this survey were based on data with proven validity. IPAQ in Korea has been proven valid in the references presented in the text. (Ref. no. 39)

Sarcopenia should not just be defined upon SMI. At least, the EWGSOP2 criteria should be used.

Author response: The diagnostic criteria for sarcopenia were based on the SMI suggested by the Asian Working Group for Sarcopenia (AWGS) rather than EWGSOP2 due to differences in physical characteristics between Westerners and Asians. 2019 AWGS guideline for sarcopenia suggested measuring muscle strength or physical performance as well as muscle mass when diagnosing sarcopenia. However, this was not measured in the KNHANES. Despite this limitation, These KNHANES data have the following strengths: the response rates and accuracy as measured by professional medical staff were high. Therefore, in future studies, it is considered meaningful to measure not only SMI but also muscle strength and body performance in a large population. This part was described as a limitation of this study (line 204-320).

  1. Materials and Methods
    Was normality tested before a chi-square or students t-test were used?
    Results
    Since the two groups are not homogenous, a comparison seems to be difficult to control for all these covariates.

Author response: Thank you for your feedback. This study is complex sample design. The data of the Korean national health and nutrition examination survey (KNHANES) used in this research is sampled by the two-stage sampling; complex sampling, and it reflects some elements such as stratification, clustering, and weights. If we analyze such data as simple random sampling, we can obtain the biased result in the variance estimates. Therefore, considering representativeness of sample and inaccurate variance estimate, we should analyze it reflecting the processing of missing value, and weights, stratification and clustering which are three elements of complex sample design. There are some differences in statistics methods which are used in simple random sampling analysis and complex sampling analysis, and in case of categorial data that is of complex sample design, it should conduct Rao-Scott test because the data does not satisfy required condition for Pearson chi-square test, and it rather increases power. So, we properly analyzed through the statistical analysis method proposed by Korean Centers for Disease Control and Prevention (KCDC). Also, the homoscedasticity and normality were satisfied in this study. In addition, in the case of this study, since the sample size is sufficiently large, this analysis method can be used even if the normality test is not satisfied.

  1. Results
    But was protein intake and calcium intake lower due to sarcopenia, or were these two leading to sarcopenia?

Author response: This study is a cross-sectional study consisting of complex sample design. Therefore, the temporal concern could not be determined, which made it impossible to accurately identify the order of fundamental causes between the two diseases. This part was described as a limitation of this study.

  1. Results
    "as age increased, the BMD levels of men and women decreased" - where is the correlation (r, p?)? How was the correlation calculated?
    "Height, weight, BMI, and WC tended to decrease as the degree of bone disorder increased." see above: where is the correlation?

Author response: We have accepted your opinion and think that we need to revise the way express the results. So, we revised it. Please, check line 171-173 in results part.

  1. Results
    For which covariates was adjusted? All? Just significant differences from table 1? Please be specific. Why show models 1-3? All covariates with significant differences in table 1 should be adjusted for.

Author response: In Table 1-3, all covariates with significant differences were adjusted. However, ASM, SMI, BMD, T-score, which are the criteria for diagnosing sarcopenia and bone disorders, and height, weight included in the BMI calculation were excluded due to multicollinearity problems.

  1. Discussion
    Again, a correlation is missing.

Author response: It is the same as the answer to question 3. The data in this study are complex sampling design, using logistic regression analysis that is most appropriate to view the association between the variables recommended by the KCDC. For this reason, other statistical analysis is not performed.
The data of the KNHANES used in this research is sampled by the two-stage sampling; complex sampling, and it reflects some elements such as stratification, clustering, and weights. If we analyze such data as pearson-correlation coefficient and simple random sampling, we can obtain the biased result in the variance estimates. The statistical analysis of this study used the most appropriate complex sampling analysis method recommended by the KCDC.
Logistic regression analysis is a regression analysis used when the dependent variable is divided into Yes or No or more categorical variables (level 1, 2, 3), and expresses the risk of the category to the dependent variable when the category 1 is a reference in an independent variable as a relative value (Odds ratio). Therefore, based on Model 4 in Table 4, the probability of sarcopenia was 2.051 times that of osteopenia and 2.258 times that of osteoporosis when bone disorder was normal. However, when classified by sex, it was 2.068 times and 3.247 times in men, respectively, but there was no significant difference in all women.

  1. Discussion
    "Third, although this study may help provide more information about the 277 nature of the relationship, it was a cross-sectional study that simultaneously measured 278 sarcopenia and bone disorder factors." - I think this is not true. You may have showed some associations, but did not provide causal connections between factors. In order to do so, a prosepective randomized trial is needed.

Author response: Thank you for your detail feedback. We have modified the way this sentence express. If my response does not match your opinion direction and must be addressed, we will do everything we can to remedy it.

Round 2

Reviewer 3 Report

I still feel that the lack of muscle strength parameters impairs the impact of your study.

The statement "The more severe the bone disorder (osteopenia, osteoporosis) group, the lower the height, weight, BMI, and WC levels compared to the normal group" still stays the same, suggesting a statistically significant correlation.

"However, ASM, SMI, BMD, T-score, which are the criteria for diagnosing sarcopenia and bone disorders, and height, weight included in the BMI calculation were excluded due to multicollinearity problems." - what were these multicollinearity problems?

Author Response

Response to Reviewer 3

Thank you for your detail review of our paper. Our answers to your points are as follows.

  1. I still feel that the lack of muscle strength parameters impairs the impact of your study.

Author response: We agree with you and think that the part you mentioned is one of the limitations of our study.

Since the data used in this study are secondary data, there was a limit to experimentally supplementing such parts. However, since this data was performed on a representative sample of the Korean adult population, the number of subjects is large, and it has the advantage of considering and adjusting many potential confounding factors.

Therefore, although parameters for muscle strength are weak, I think this study is significant because accurate muscle mass and osteoporosis diagnostic data measured by DEXA and various confounding variables are considered.

  1. The statement "The more severe the bone disorder (osteopenia, osteoporosis) group, the lower the height, weight, BMI, and WC levels compared to the normal group" still stays the same, suggesting a statistically significant correlation.

Author response: Thank you for your review. We revised it. Please, check again line 172-173.

  1. "However, ASM, SMI, BMD, T-score, which are the criteria for diagnosing sarcopenia and bone disorders, and height, weight included in the BMI calculation were excluded due to multicollinearity problems." - what were these multicollinearity problems?

Author response: Multivariate regression analysis is a regression analysis used when the dependent variable is divided into Yes or No or more categorical variables (level 1, 2, 3), and expresses the risk of the category to the dependent variable when category 1 is a reference in an independent variable as a relative value (Odds ratio).

The independent and dependent variables used in the multiple regression analysis of this study are sarcopenia and bone disorders (bone reduction, osteoporosis). When performing multiple regression analysis, ASM, SMI, BMD, and T-score, which are diagnostic criteria for sarcopenia and bone disorders, cannot be used as confounding variables. Therefore, in Table 4, these variables were not initially considered because they were not confounding variables.

Also, Covariates that are used to analyze and interpret clinical trial data can become confounding factors. They can be statistically controlled for during analysis, which results in a more direct measurement of the relationship between the variables of interest.

Therefore, this study attempted to see if there was a association between sarcopenia and bone disorder even when many variables shown in Table 1-3 were included as confounding variables. However, when applying these confounding variables, independence between confounding variables must be guaranteed due to the problem of multicollinearity. Otherwise, the association between sarcopenia and bone disorder can be underestimated or overestimated.

Therefore, the degree of multicollinearity must be verified to ensure the independence of confounding variables before conducting multiple regression analysis. To this end, the Variation Inflation Factor (VIF) method was used. Height, weight, and BMI are variables that are significantly related to each other, so the VIF value was very large. For this reason, BMI was used as a confounding variable instead of height and weight. In other variables, the VIF value was less than 10, so it was used in the analysis.

In addition, the reason for classifying it into Model 1-4 in Table 4 is that the forward selection method was used for confounding variables to be included in the regression model. Please check the paper below that presents a table of similar results to this study using KNHANES data.

  • Shin, D., Joh, H. K., Kim, K. H., & Park, S. M. (2013). Benefits of potassium intake on metabolic syndrome: The fourth Korean National Health and Nutrition Examination Survey (KNHANES IV). Atherosclerosis, 230(1), 80-85.
  • Yun, S., Nguyen, H. D., Park, J. S., Oh, C., & Kim, M. S. (2021). The association between the metabolic syndrome and iron status in pre-and postmenopausal women: Korean National Health and Nutrition Examination Survey (KNHANES) in 2012. British Journal of Nutrition, 1-11.
  • Ki, E. Y., Do Han, K., & Park, Y. G. (2017). Relationship between duration of breast-feeding and obesity in Korean women: The korea national health and nutrition examination survey (KNHANES) 2010–2012. Maturitas, 102, 41-45.